# Cytokines Meet Phages: A Revolutionary Pathway to Modulating Immunity and Microbial Balance

**DOI:** 10.3390/biomedicines13051202

**Published:** 2025-05-15

**Authors:** Rossella Cianci, Mario Caldarelli, Paola Brani, Annalisa Bosi, Alessandra Ponti, Cristina Giaroni, Andreina Baj

**Affiliations:** 1Department of Translational Medicine and Surgery, Catholic University of Sacred Heart, 00168 Rome, Italy; mario.caldarelli01@icatt.it; 2Fondazione Policlinico Universitario A. Gemelli, Istituto di Ricerca e Cura a Carattere Scientifico (IRCCS), 00168 Rome, Italy; 3Department of Medicine and Technological Innovation, University of Insubria, 21100 Varese, Italy; pbrani@uninsubria.it (P.B.); annalisa.bosi@uninsubria.it (A.B.); aponti@uninsubria.it (A.P.); cristina.giaroni@uninsubria.it (C.G.); andreina.baj@uninsubria.it (A.B.); 4Laboratory of Microbiology, ASST Sette Laghi, 21100 Varese, Italy

**Keywords:** bacteriophage, cytokines, immune system, cancer, gut microbiota

## Abstract

Bacteriophages are a unique and fascinating group of viruses, known for their highly specific ability to infect and replicate within bacterial cells. While their potential as antibacterial agents has been recognized for decades, recent research has revealed complex interactions between phages and the human immune system, offering new insights into their role in immune modulation. New evidence reveals a dynamic and intricate relationship between phages and cytokines, suggesting their ability to regulate inflammation, immune tolerance, and host–pathogen interaction. Herein, we review how phages affect the production of cytokines and the behavior of immune cells indirectly by lysis of bacterium or directly on mammalian cells. Phages have been shown to induce both pro- and anti-inflammatory responses and recently, they have been explored in personalized immunotherapy, cancer immunotherapy, and microbiome modulation, which are the focus of this review. Several challenges remain despite significant progress, including practical obstructions related to endotoxins along with host microbiome variability and regulatory issues. Nevertheless, the potential of bacteriophages to modulate immune responses makes them attractive candidates for the future of precision medicine.

## 1. Introduction

Bacteriophages, meaning “bacteria eater”, were first discovered during the infancy of microbiology by Félix d’Hérelle in 1917. His pivotal discovery came after observing that a filtrate of a dysentery patient’s stool had the ability to lyse and kill specific bacterial strains. This phenomenon prompted him to theorize the existence of viruses that infect bacteria, which he later isolated and characterized extensively [1].

Despite a long period in which phage therapy was largely overlooked, the rise in antibiotic resistance in the late 20th century reignited interest in these viruses. The emergence of multidrug-resistant bacteria has led to an urgent need for alternative treatment strategies, prompting researchers to revisit and rigorously study phage therapy as a viable option for combating bacterial infections that do not respond to traditional antibiotics [2]. An example of this resurgence of interest is the emergence of personalized phage therapy in clinical settings, where specific phages are selected based on the susceptibility profile of pathogenic bacteria isolated from patients. This targeted approach has shown promising results in the treatment of complex infections, such as those associated with cystic fibrosis and chronic wounds [3].

At the same time, contemporary technologies have enabled significant advancements in phage research, allowing for genome sequencing of various phages, providing insights into their structure, diversity, and functionality [4]. Today, scientists manipulate phage genomes to enhance their efficacy and host range, paving the way for tailored therapeutic interventions [5].

The potential of phages, not only as therapeutic agents, but also as tools for understanding bacterial genetics and ecology has been explored. For instance, the use of phages in the development of recombinant DNA technology, revolutionized research in life sciences [6]. The introduction of phage display technology in the 1990s allowed scientists to identify and develop novel proteins and peptides for various applications, including drug development and diagnostics [7].

Today, the applications of phages span from phage therapy and vaccine development to their engineering for use in precision medicine [8]. Also, there is great promise for phage engineering, with the aim to target specific pathogens while minimizing impacts on beneficial microbiota [9].

As researchers continue to unravel the complexities of phage biology and their interactions with the immune system, the potential of these versatile viruses to serve as a cornerstone of future medical therapies becomes increasingly evident [10].

Furthermore, phages are being explored for use in biocontrol, reducing foodborne pathogens in agricultural settings and enhancing the safety of food production. The integration of phages into biotechnological applications is transforming numerous industries, from bioremediation of contaminated environments to the development of phage-based biosensors [11].

Surprisingly, the growing understanding of the human microbiome has further underscored the significance of phages in shaping microbial communities and regulating immune responses, revealing their importance beyond mere bacterial predation [12].

In fact, while traditionally considered inert to the mammalian immune system, mounting evidence suggests that they may be capable of modulating immune responses. Phages can be detected by immune cells and influence cytokine secretion, either directly or indirectly, raising questions about their role in inflammation, immune tolerance, and cancer immunotherapy. Interest has been garnered in leveraging them as innovative immunotherapeutic agents [13].

The aim of this narrative review is to highlight the intricate relationship among phages, microbiota, and the immune system to shed light on potential therapeutic applications in both inflammatory and infectious diseases and on developing phage-based therapies. For this reason, we have extensively reviewed the scientific literature, giving priority to high-impact scientific journals and recent studies of the last decade, including randomized controlled trials, meta-analyses, and reviews. The keywords we used were ‘bacteriophage’, ‘immune modulation’, ‘microbiota’, and ‘cancer’.

## 2. Mechanisms of Immune Modulation

The advent of high-throughput next-generation sequencing (NGS) technologies has enabled us to analyze complex and diverse microbial ecosystems that are otherwise difficult to culture. This advancement has also enhanced our understanding of the viral component of the microbiome, known as the *phageome* [14]. Historically, bacteriophages (phages) were thought to influence the immune system only indirectly—primarily through their interactions with bacterial hosts [15]. However, recent research suggests a much more intricate relationship (Figure 1).

Although phages do not directly infect mammalian cells, they can still elicit immune responses [16], depending on the type of phage, and the specific environment. In some instances, phages promote anti-inflammatory effects; in others, they stimulate pro-inflammatory pathways [17], highlighting a range of interactions with the human immune system that remain only partially understood.

Indeed, recent findings suggest that bacteriophages have the capacity to interact with the host immune system in both direct and indirect manners. They function as ligands for pattern recognition receptors (PRRs), including TLR3, TLR7, and TLR9, and NOD-like receptors (NLRs), as well as cytoplasmic sensors such as cGAS and AIM2.

Phage DNA, particularly ssDNA and dsDNA phages, is recognized by endosomal TLR9 and cytosolic sensors, leading to the activation of MyD88- or STING-dependent pathways and the production of cytokines such as IFN-α, IL-6, IL-12, and IL-10 [18]. Filamentous phages such as M13 have the capacity to persist within mucosal tissues, where they are internalized by epithelial cells and antigen-presenting cells (APCs). This process modulates downstream immune signaling and antigen presentation [19]. Research has indicated that modulation of T-cell responses, encompassing the expansion of Th1 and Th17 subsets, is contingent on phage structure, host background, and exposure history [20]. Due to phage-derived lysis of bacterial populations, PAMPs (e.g., LPS, peptidoglycans) are also released. These PAMPs activate TLR2/TLR4 and NOD-like receptors, thereby reinforcing innate immune activation [21]. Amy et al. reported that RNA phages from the Emesvirus genus trigger specific immune responses in their bacterial hosts. The phage MS2 activates a bacterial immune protein [18], called bNACHT25 and related to mammalian NLRs. The protein uses a NACHT domain to start defense mechanisms. The phage’s coat protein (CP) drives the activation. There is no direct interaction between CP and bNACHT25. The host chaperone DnaJ is required for the detection process. Phages are not recognized through direct binding but can still disrupt host cellular processes.

When these receptors recognize phage particles, they activate intracellular signaling cascades that alter cytokine production and influence downstream immune activity [22].

Emerging evidence suggests that phages entering a host cell or indirectly stimulating intracellular immune sensors may activate various innate immune pathways. Originally characterized in a study on *Listeria monocytogenes*, there are two distinct immune pathways: a vacuolar response that is entirely Myeloid differentiation primary response (MyD) 88-dependent and drives expression of pro-inflammatory cytokines such as Interlukine (IL)1α, IL1β, and tumor necrosis factor (TNF), and a cytosolic response that is Interferon Regulatory Factor 3 (IRF3)-dependent and activates a distinct set of 106 genes, including Type I interferons, such as IFNβ. Intriguingly, only seven of those genes were induced directly and solely in infected cells without secondary cytokine signaling, and among them are IFNβ, MYD116, and members of the Interferon Induced Protein with Tetratricopeptide Repeats family [23,24].

These findings indicate that phages entering the cytosol, either naturally or via delivery systems, could engage similar PRR pathways. Specifically, phage DNA might activate IRF3 and NFκB via cytosolic DNA sensors, while structural components or peptidoglycan residues associated with phage particles could elicit NOD2 recognition. Joint signaling through NOD2 and TLR at the level of NFκB nuclear abundance exemplifies an elaborate mechanism of immune modulation where intracellular phages might influence Type I interferons’ response through coordinated engagement of multiple PRRs [25].

Moreover, experiments on NOD2-deficient macrophages indicated diminished IFNβ induction in intracellular bacterial infections when TLR signaling was absent, suggesting that the NOD2-mediated amplification of interferon responses may also be relevant for other intracellular pathogens, thus including phages in their interactions with hosts. This information supports a model suggesting that phages may act not only as immunogens but also as modulators of the host’s innate signaling via classical bacterial infection pathways [26]. Additionally, when phages lyse bacterial cells, they can release bacterial components such as lipopolysaccharides (LPS) and peptidoglycans—known as pathogen-associated molecular patterns (PAMPs)—which further drive cytokine-mediated immune responses. Depending on the bacterial species involved, these immune responses may skew either toward inflammation or resolution [27].

Phages can also stimulate IFN responses. Both type I interferons (IFN-α, IFN-β), which play roles in antigen presentation and suppression of macrophage activity, and type II interferons (IFN-γ), which drive Th1-mediated inflammation, may be induced by phage exposure [28]. For example, Mori et al. demonstrated that DNA from the filamentous M13 phage triggered serum IFN release in mice, providing protection against vaccinia virus lesions [29]. Similarly, the E. coli filamentous phage has been shown to be immunogenic in mice, activating TLR9 via its single-stranded DNA, in a manner comparable to eukaryotic viruses [30,31].

Notably, phages can exert differing effects on cytokine profiles depending on their type and the context of infection. The immunological effects of phages become even more complex when considering pleiotropic cytokines such as IL-6 and IL-10 [32]. IL-6 is primarily pro-inflammatory and promotes Th2 responses, whereas IL-10 plays a regulatory role in controlling inflammation and protecting tissues from damage. Exposure to phages can induce various immune responses depending on their genome. Peripheral blood mononuclear cells (PBMCs) exposed to double-stranded DNA (dsDNA) phages produce pro-inflammatory cytokines (TNF, IL-1α, IL-1β, IL-6), alongside IL-10. On the other hand, double-stranded RNA (dsRNA) phages elicit the production of IL-6, IFN-β, and CCL4 [32]. Furthermore, phage strain specificity plays a role—lung epithelial cells exposed to podovirus PEV2 show increased IL-6 and TNF production, while exposure to siphovirus DMS3 does not induce a similar response [33].

Some phages have been shown to reduce pro-inflammatory cytokines such as IL-6, TNF-α, and IL-1β, thus alleviating systemic inflammation in diseases like inflammatory bowel disease (IBD) and rheumatoid arthritis [34]. On the other hand, other phages can promote the release of these cytokines, particularly during bacterial lysis, thereby enhancing acute inflammatory responses [15]. Additionally, phages can modulate anti-inflammatory mediators like IL-10 and TGF-β, helping to prevent excessive inflammation and autoimmunity, which contributes to maintaining immune homeostasis.

These findings underscore the complex and dualistic nature of phage–immune system interactions and their role in fine-tuning immune responses [21]. A key observation is that phage activity is highly context-dependent, varying across different tissues, host conditions, and immune microenvironments.

Gogokhia et al. provided further evidence of phage-driven immune activation by showing that dsDNA phages from *Lactobacillus*, *Escherichia*, or *Bacteroides* induced colitis in germ-free mice via TLR9-mediated responses, promoting IFN-γ-producing Th1 cells [35]. Importantly, in human studies, patients with ulcerative colitis (UC) who responded to fecal microbiota transplantation (FMT) exhibited lower levels of phages compared to non-responders, suggesting that phages might contribute to chronic intestinal inflammation [35]. Phage preparations from UC patients also triggered greater IFN-γ production than those from healthy donors, further supporting this hypothesis [36].

Phage-induced immune responses include both humoral and cellular components and are influenced by factors such as phage structure, route of administration, and host immune status [37]. While mice consistently produce antibodies against phages, human responses appear more variable. Oral phage therapy typically elicits minimal antibody production, whereas intravenous administration may result in varying levels of IgG and IgM antiphage antibodies [38]. This immune recognition is particularly relevant in the context of phage therapy, as it can diminish therapeutic efficacy [39].

Once in the body, the bacteriophages are absorbed and metabolized by antigen-presenting cells (APCs) which degrade and process the antigens in peptides. They are expressed using the MHC class I or II pathways [40]. MHC class-II presentation activates CD4^+^ helper T cells that in turn help B cells to produce antibodies against tumor cells. Such an antibody response, in turn, may result in tumor destruction through complement activation, natural killer cells, or macrophage phagocytosis. MHC class I mediated presentation stimulates CD8^+^ cytotoxic T cells to directly kill target cells, including tumor cells by releasing cytotoxic proteins, such as granzyme and perforin [41].

In conclusion, while phages clearly impact adaptive immunity, their immunogenicity varies considerably. Further studies are essential to fully understand how phages influence immune regulation and to harness their therapeutic potential [42].

## 3. Direct Interactions with Human Cells

Phages are an integral part of the human microbiome and colonize virtually all body niches, resulting in continuous exposure to a diverse phage community [43]. This is especially evident in the gut, which hosts a highly diverse microbial ecosystem [44]. In this environment, phages play a key role in shaping and diversifying the gut microbiome throughout life [45].

While phages are well known for influencing bacterial populations, emerging evidence suggests they also interact directly with mammalian cells underlying the microbiota [46,47]. However, the precise nature of these interactions—how phages engage with mammalian cells, modulate innate immune responses, and potentially affect downstream cellular functions—remains incompletely understood.

The internalization of phages can occur via non-specific endocytosis, encompassing macropinocytosis and receptor-mediated pathways such as clathrin-mediated endocytosis. Filamentous phages, such as M13, have the capacity to bind to specific receptors on epithelial cells, resulting in internalization and modulation of signaling pathways. This process includes the activation of TLR9 when the phages are transported to endosomal compartments [48]. Once inside the cell, phages may localize to the cytoplasm or endosomal vesicles, where they can either be degraded or interact with intracellular receptors. Research has indicated that phages such as T4 and T7 have been observed to elicit anti-inflammatory effects, while Pf phage, linked to *Pseudomonas* infections, has been shown to stimulate pro-inflammatory responses through TLR activation [49]. Intriguingly, internalized phages have been observed to traverse epithelial barriers via transcytosis, thereby preserving their structural integrity and achieving systemic circulation without substantial degradation [50]. Recent data suggest that phages localized near the nucleus in MAC-T cells may influence nuclear functions, although replication within mammalian cells has not been confirmed [51]. The capacity to traverse epithelial layers may be influenced by factors such as phage morphology, cell type and environmental conditions, as evidenced by studies examining oral administration and systemic distribution [22].

It has been shown that mammalian cells can internalize phages through various mechanisms, leading to their accumulation in active form within the cells [52]. The most common entry route is non-specific uptake via macropinocytosis [43], resulting in phage localization within intracellular macropinosomes [47]. Once internalized, phages can elicit a wide range of immune responses, including both anti-inflammatory [53] and pro-inflammatory effects [19].

For example, Bichet et al. demonstrated that highly purified T4 bacteriophages, a species of bacteriophages that infect *Escherichia coli*, can interact directly with mammalian cells, causing substantial alterations in cells without triggering inflammatory DNA sensing pathways [50]. T4 phages activated Protein Kinase B signaling upon internalization via macropinocytosis, leading to enhanced cell metabolism, survival, and cytoskeletal reorganization. Furthermore, phage treatments downregulated Cyclin-dependent kinase 1, resulting in a slowdown of cell cycle progression while promoting an extension of the G1 phase for growth. These results suggest that mammalian cells may view phages not as immune threats but rather as signals or resources meant to contribute to supporting cellular growth and homeostasis.

Interestingly, the nature of these responses appears to depend on the specific phage involved. For instance, phages T4 and T7 are associated with anti-inflammatory activity, while the filamentous Pf phage has been linked to pro-inflammatory effects [48].

Some phages can induce receptor-mediated endocytosis, by binding to specific mammalian receptors [54]. The M13 phage, for example, can be internalized by enterocytes and endothelial cells, and this process is inhibited by chloroquine—suggesting the involvement of a clathrin-mediated endocytic pathway [55,56].

Zamora et al. analyzed how lytic bacteriophages employed in phage therapy interact with human airway epithelial cells (AECs) in the context of cystic fibrosis [57]. Although phages can achieve limited internalization in AECs, these cells react to the exposure to phages by altering their gene expression and releasing antiviral and pro-inflammatory cytokines, with varying responses depending on phage types and environmental conditions. These results point to a potential role of phages in influencing immunity within the host epithelium, suggesting that selection of the phage for therapy should not only involve bacterial targeting but also include consideration of the host cellular responses.

Phages are in the epithelial mucus layer near the epithelial cells. While bacterial translocation across the epithelium is a well-documented contributor to infection [58], much less is known about phage translocation. It likely depends on multiple factors, as the oral administration of phages has led to both successful [59,60] and unsuccessful [61,62] systemic distribution [63].

Notably, phages can traverse epithelial barriers via a non-specific transcytosis mechanism, preferentially moving in an apical-to-basal direction across various epithelial cell types and phage morphologies [52]. The epithelial cells can assume phage particles and, once inside, phages pass through the Golgi apparatus and are then exocytosed at the basal side and reach the blood.

Conversely, other studies have reported that phagocytosed phages may accumulate near the nucleus of MAC-T cells [64]. The presence of phages near or within the nucleus raises intriguing questions: could phages replicate or translate their genome in mammalian cells? Could phage-derived RNA trigger cellular responses, or might nuclear-localized phages influence host cell functions?

## 4. Drivers of Microbiota Composition

Following years of intensive research on the microbiome, it is now increasingly evident that phages play a central role in shaping bacterial populations within the human gut. Found in virtually every microbial environment, the interactions between phages and their bacterial hosts are complex and multifaceted.

It is evident that phages play a pivotal role in the shaping of the composition of the gut microbiota through dynamic predator–prey interactions. The ‘kill-the-winner’ model demonstrates how phages target the most dominant bacterial strains, thereby enabling less abundant species to persist, and thus maintaining microbial diversity [59,65]. Furthermore, lysogenic phages integrate into bacterial genomes as prophages, promoting bacterial fitness through the acquisition of beneficial genes, including antibiotic resistance and virulence factors [16,60,61,66]. This phenomenon has the capacity to facilitate bacterial adaptation to environmental stressors, including the presence of antibiotics or immune pressure.

One of the approaches to study phage-bacterial interactions in the gut is the identification of the most specific phage–host pairs. Recently, different in silico methods have been described that aim to predict possible interactions [67]. Sequence similarity or machine-learning models can be applied for the prediction of receptor–ligand binding, prophage integration, and protein–protein interactions between the phages and their co-isolated bacterial hosts. A simpler approach is that of analyzing CRISPR spacer content for the identification of phage–host relationships. The high-throughput metagenomic metaHi-C technique entails chemical cross-linking to physically link co-habiting phage and bacterial genomes followed by sequencing which provides high-resolution data on chromosomal interactions between phages and their bacterial hosts [68].

Importantly, phages do not act in isolation but rather participate in tripartite symbioses involving the virus, bacteria, and human host [65].

These dynamic interactions contribute to the rapid coevolution of phages and bacteria, shaping the structure of both communities [69]. The presence of phage-resistance genes in gut bacteria [70] and the detection of CRISPR spacers targeting phage sequences [71] reflect ongoing evolutionary pressure. Bacteria can progress phage resistance through changes in surface receptors to avoid phage infection [72].

The predator–prey relationship between phages and bacteria is further influenced by the intestinal environment and spatial localization within the gut [73] with phages exhibiting higher concentrations in the intestinal lumen compared to mucosal surfaces, where mucus-adherent bacteria may evade lytic phage activity [74]. It has been hypothesized that phages adhering to mucins via Ig-like domains can establish a localized niche that protects against bacterial invasion [75]. As demonstrated by Howard-Varona et al., the decisions made by temperate phages regarding the lytic or lysogenic cycle are significantly influenced by environmental conditions, including pH, nutrient availability, and host immune responses [76].

For example, phages are more abundant in the intestinal lumen than in the mucosa, which may serve as a refuge for bacteria [77]. Interestingly, phages can also adhere to the mucus layer via immunoglobulin (Ig)-like domains on capsid proteins such as Hoc. This interaction, described by the Bacteriophage Adherence to Mucus (BAM) model, suggests a protective role for phages against bacterial invasion [74].

The efficiency of phage infection is also location dependent. For instance, *E. coli* is more susceptible to phage infection in the ileum than in the colon, where phage replication is also more active than in the cecum [78]. In this context, phage-mediated DNA transduction can support bacterial metabolism or transfer virulence factors, such as the Shiga toxin–encoding prophages in *E. coli* [79,80].

Switching between lysogenic and lytic cycles is a tightly regulated process often triggered by stress signals. The maintenance of lysogeny requires ongoing expression of repressor proteins that suppress lytic genes. When these repressors are reduced—such as through DNA damage—the prophage is excised, initiating the lytic cycle [76].

Temperate phages, for example, can produce a pro-lysogenic signaling molecule called “arbitrium”, which helps guide subsequent infections toward either lysogeny or lysis depending on population context [81]. Initially, entering the lytic cycle may maximize phage replication [82], but as infection spreads, lysogeny often becomes the preferred strategy, supporting the long-term survival of both phage and host bacteria. This decision-making process allows phages to respond to host density and lysogen abundance, optimizing their replication strategy [83].

This balance is further refined through coordinated decisions among phages infecting the same host. Many prophages confer “superinfection exclusion”, providing their lysogenic hosts with immunity against further phage infections [84]. In some cases, such as with phages carrying CRISPR-Cas systems, these mechanisms actively target other phages, enhancing the competitive advantage of lysogens [85].

For *Vibrio* phage VP882, the choice between lysogeny and lysis is influenced by bacterial quorum-sensing molecules. VP882 encodes a homolog of a host regulatory protein that reduce the expression of lytic genes, thus modifying the host virulence [86].

Bacterial metabolism can also impact phage fitness. Metabolic signals, such as short-chain fatty acids (SCFAs), may inhibit bacterial pathways targeted by phages or activate prophage expression [87]. In turn, phages can adapt to these signals by altering their own metabolic output to better target susceptible bacterial strains [88]. This evolutionary “arms race” unfolds across molecular, cellular, and organismal levels.

The gut microenvironment also modulates phage behavior. Inflammatory conditions, for instance, have been shown to promote the activation of temperate phages and the spread of virulence genes, as observed in *Salmonella enterica* serovar Typhimurium during intestinal inflammation [89].

A very new and on-the-way approach to assessing the causal role of phages in gut health is the use of FMT [90]. Traditionally application of bacteriophages has been on cell-free fecal filtrates with a high concentration of microbes and inhibitors to restore health from *Clostridioides difficile* infections, thus implicating phages as therapeutic agents. Some studies showed that donor Caudoviricetes phages persisted in the host for more than a year, and successful engraftment was related to the positive FMT outcome [91]. In mouse models, the transfer of fecal virome improved symptoms of type II diabetes and stress responses, suggesting an action either by selectively decreasing harmful bacteria or by modulating immunity in the host [92]. Importantly, phages from *Pseudomonas aeruginosa* have been shown to activate mammalian type I interferon pathways, restraining pro-inflammatory TNF signaling, and allowing continuing bacterial infection [35]. New acquisitions on the immunomodulatory effects of phages and the future development of phage-based therapeutics that would directly reshape the microbiota or engage the human immune system were part of these discoveries.

Given these intricate interactions, the idea of using phages as therapeutic tools has attracted renewed interest. Importantly, phage therapy aims to harness these dynamic interactions to restore microbial homeostasis, either by selectively depleting pathogenic bacteria or by introducing beneficial prophages that counteract dysbiosis [21].

## 5. Phages and Cancer

Cancer represents an important health problem. In recent years, innovative use of bacteriophages has been considered in this field. Phage-based cancer therapy offers potential minor side effects with consequent better patient responses.

Several techniques are used to enable phages to recognize tumor cells. One is phage display technology, which has proven highly effective for identifying peptides or antibodies that bind to specific target molecules. Libraries of peptides and antibodies created through phage display are employed to isolate phage clones, peptides, or antibodies with strong affinity for tumor cells. Additionally, phage proteins can be chemically modified with various dyes, allowing the development of phage-based probes to visualize cancer cells [93]. Furthermore, phages can be genetically engineered to express peptides or antibodies that specifically target tumor-associated antigens, such as EGFR or GRP78, which are often overexpressed in various types of cancer. For example, a study by Gaj et al. reports a targeted photodynamic therapy (PDT) of EGFR overexpressing ovarian cancer using a M13 bacteriophage (M13r), genetically engineered to expose an EGF receptor binding peptide. The phage was coupled with the photosensitizer chlorin e6 (Ce6) to make M13r-Ce6. Once illuminated, M13r-Ce6 was able to produce reactive oxygen species and thus efficiently kill SKOV3 and COV362 ovarian cancer cells, even at doses where Ce6 was ineffective. The treatment was effective in terms of cellular uptake, mitochondrial localization, downregulation of EGFR, and induction of autophagy. This phage-mediated approach represents a highly promising targeted PDT strategy for ovarian cancer [94]. Moreover, scientists have isolated from a phage display library an antibody fragment, that is capable of specifically attaching to cancer cells by recognizing the surface-located GRP78 protein. It is reactive against several cancer types, such as breast, lung, and melanoma, and at the same time, it shows low binding to normal tissues. The limited presence of GRP78 on the surface of tumor cells makes it an interesting target in the field of cancer-related antibody research and treatment since it could potentially have a significant effect on the fight against cancer [95], with the potential for advancing to the next stage in the development of cancer immunotherapy. This allows phages to selectively infect cancer cells while sparing healthy ones, thereby avoiding many of the adverse effects commonly associated with chemotherapy, such as nausea, vomiting, alopecia, anorexia, and bone marrow suppression.

Additionally, because phages exhibit single-hit kinetics (i.e., one phage infects one cell), lower quantities are required compared to chemical therapies. Phages also offer practical advantages: they are stable across a wide pH range, inexpensive to produce, have a long shelf life, and are more environmentally friendly to dispose of than traditional chemotherapeutics [96].

Phage-based therapies work through two main mechanisms: by stimulating immune responses or by serving as delivery vehicles for therapeutic agents. In the first mode, phages activate the immune system to detect and destroy cancer cells [96]. Strengthening the immune system in rapidly identifying cancer-specific markers can avoid metastatization or recurrence [97].

Phages can represent an emerging strategy for vaccine development due to the ability to present tumor-specific antigens. Once internalized, phages are degraded by antigen-presenting cells (APCs), and their antigens are processed and presented on major histocompatibility complex (MHC) molecules [40]. MHC class II presentation leads to CD4^+^ T cell activation, which in turn promotes B cell activation and antibody production, contributing to tumor cell elimination. In contrast, MHC class I presentation activates CD8^+^ cytotoxic T cells, which release granzyme and perforin to induce apoptosis in tumor cells [41].

Phages can be carriers for therapeutic molecules. Gene therapy—introducing engineered material into cells to modify their function—is a major emerging area. However, targeting inefficiencies, vector instability, and potential infection risks limit the use of eukaryotic viral vectors [98]. In contrast, phages are highly specific, stable, and incapable of infecting mammalian cells [99]. Chimeric phages engineered to target cancer-specific receptors and encode eukaryotic transgenes have been developed to combine the strengths of both systems [100].

Phages’ unique properties—including nanoscale size, non-pathogenicity, polyvalent surfaces, and modifiability—make them excellent candidates for therapeutic and diagnostic use in solid tumors [101]. Their uniformity and ease of production using bacterial hosts further increase their appeal [102]. The phage capsid serves as a robust platform for conjugating diverse molecules, such as drugs or imaging probes, thus allowing for controlled and targeted delivery [103].

It is possible to use phages to increase drug penetrance into cancer tissues [104]. This property also helps them evade rapid clearance by organs like the liver, spleen, and kidneys, thus prolonging their circulation time and enhancing therapeutic impact [105]. Engineered phages can also be equipped with drugs, targeting ligands, or therapeutic agents tailored to specific treatment strategies or individual patient needs [106].

Phages can cross barriers, such as the blood–brain barrier, crucial for treating brain tumors [107], and the fibrotic microenvironment of pancreatic cancers [108], and are able to enhance therapeutic efficacy [109].

One compelling example is the development of a bacteriophage-derived particle (PDP) carrying a transgene encoding sTRAIL, a cytokine that induces cancer cell apoptosis. This PDP showed potent tumor growth inhibition in chondrosarcoma models [110].

Phage display technology has two primary applications: the presentation of antigens on phage surfaces and the identification of novel antigens [111]. This method has been instrumental in discovering tumor-specific surface markers and developing anticancer peptides. Phage-displayed vaccines allow the fusion of tumor antigens with phage proteins to generate highly immunogenic constructs [112]. Various tumor antigens—including VEGFR2, EGFR, HER2, MAGE, MUC1, FGFR, Flt4, and tumor-associated carbohydrate antigens—have been used in preclinical vaccine candidates [113].

There is a growing interest in targeted drug delivery to improve chemotherapy efficacy and reduce its side effects. Phage display-derived peptides can home on tumor cells, guiding drug molecules directly to the site of disease. Studies have demonstrated that phages can enhance the effects of chemotherapy while minimizing toxicity [114].

Phage libraries, such as those based on M13, have been used to identify peptides capable of crossing cell membranes and transporting drugs into cancer cells—examples include LTVSPWY and WNLPWYYSVSPT [115]. Another example is a dual-conjugate consisting of doxorubicin and a peptide from the 12-mer M13 library (AGKGTPSLETTP), which exhibited strong anticancer activity in hepatocellular carcinoma models [111].

Furthermore, M13 phages modified with an anti-PSMA antibody and conjugated to single-walled carbon nanotubes have been used for prostate cancer imaging [116]. Similarly, M13 phages that mimic epithelial growth factors have been used to deliver siRNA to lung cancer cells [117].

Phages have also shown promise in photodynamic therapy (PDT), a treatment that uses light to activate photosensitizers. For example, MS2 and M13 phages labeled with aptamers targeting breast cancer cells have been used to deliver photosensitizers, leading to selective cancer cell destruction [118,119]. In mouse models, M13 phages showed to ameliorate survival in colorectal cancer [120]. Recent advancements include dual-functional phages that combine imaging capabilities with therapeutic functions, such as M13 phages conjugated with single-walled carbon nanotubes for targeted prostate cancer imaging [51].

In conclusion, phage display technology represents an innovative and versatile platform in tumor immunology, applicable both as a vaccine strategy and therapeutic agent. Engineered phages offer a compelling alternative to eukaryotic virus-based vectors for cancer diagnostics and gene therapy. However, before clinical application, further research is needed to better recognize phage–host immune interactions, potential mechanisms of resistance, hypersensitivity reactions, and long-term safety.

## 6. Therapeutic Applications

The application of bacteriophages in immunomodulatory therapeutics represents a revolutionary shift in the way immune responses are managed. By leveraging the natural specificity and genetic versatility of phages, researchers have been able to develop phage-based therapies that precisely manipulate immune responses. This approach is particularly valuable in diseases characterized by immune dysregulation, such as autoimmune disorders, chronic inflammation, and cancer [22].

The rapidly advancing understanding of bacteriophage interactions with host immunity is unveiling transformative opportunities in immunomodulatory therapeutics. One of the most intriguing developments in this field lies in the interplay between phages and cytokines, critical signaling molecules that orchestrate immune responses (Figure 2). This emerging knowledge not only enhances our grasp of phage biology but also positions bacteriophages as powerful agents capable of influencing host immune landscapes across a wide array of diseases.

Engineered phages offer a particularly promising platform in this context. Through genetic modification, phages can be designed to deliver or express specific immunomodulatory molecules, enabling targeted and precise manipulation of cytokine responses. This level of control opens the door to disease-specific immune tuning, where phages might be used to reduce inflammation in autoimmune conditions, augment immune responses to persistent infections, or even enhance antitumor immunity. Their natural specificity and biological safety, combined with their capacity for genetic customization, make them uniquely suited for this kind of precision immunotherapy [121].

In addition to engineered constructs, there is growing interest in bioactive peptides derived from phages that can modulate cytokine pathways. These peptides have the potential to serve as novel therapeutic agents, particularly in complex or treatment-resistant diseases. When used alongside conventional treatments, such peptides could enhance therapeutic efficacy, support immune system balance, and reduce the side effects associated with broad-spectrum immunosuppressants. As adjunct therapies, they could play a crucial role in restoring immune equilibrium in conditions where immune dysregulation plays a central role [122].

Phages can be tailored to express anti-inflammatory cytokines like IL-10 or TGF-β, which help reduce excessive immune activation, or pro-inflammatory cytokines like IFN-γ to enhance antiviral or antitumor responses [13]. The dual role of phages as both immune stimulants and delivery vehicles makes them highly adaptable for personalized medicine.

The field is also steadily moving toward personalized phage therapy, recognizing the considerable variability in human microbiomes and immune responses. Since cytokine profiles differ between individuals, tailoring phage therapies to align with each patient’s unique immunological and microbial landscape could maximize therapeutic effectiveness. Personalized phage interventions may prove especially useful in chronic inflammatory and immune-mediated disorders, where standard approaches often fail to deliver consistent results. By leveraging individual microbiome compositions and immune reactivity patterns, researchers can begin to design more adaptive, responsive treatments that reflect the complexity of each patient’s biology [123].

In the context of oncology, phages have been engineered to present tumor antigens on their surfaces, thereby enhancing the activation of cytotoxic T lymphocytes (CTLs) and promoting robust antitumor immunity [124]. This strategy not only targets the tumor cells directly but also primes the immune system to recognize and destroy metastatic cells, reducing the likelihood of recurrence. Moreover, the use of phages in conjunction with immune checkpoint inhibitors has shown potential in preclinical models, where the combination leads to improved survival and decreased tumor burden [124].

Yet, as with any biologic agent, the path toward clinical application is not without challenges. One of the most significant concerns involves endotoxins—particularly lipopolysaccharides (LPS)—that may be present in phage preparations. LPS, a component of the outer membrane of Gram-negative bacteria, is a potent activator of innate immunity and can provoke unwanted inflammatory responses. When phages lyse Gram-negative bacteria, large amounts of LPS may be released, exacerbating inflammation and complicating the immune response to both the bacteria and the therapeutic phages themselves [39].

Interestingly, this inflammatory cascade can have paradoxical effects. On one hand, the heightened immune response may accelerate bacterial clearance; on the other hand, it may reduce the persistence and therapeutic window of phages. Immune activation through endotoxins may lead to the rapid production of anti-phage antibodies, which neutralize the phages and limit their bioavailability. Experimental data from murine models have shown that LPS-induced inflammation can significantly reduce phage titers in organs, such as the liver, spleen, and kidneys, in part due to enhanced clearance by phage-specific antibodies [15,125].

Furthermore, the host’s pre-existing inflammatory state also appears to influence phage pharmacokinetics and dynamics. This suggests a complex feedback mechanism in which the immune system not only responds to phages and their bacterial targets but also actively shapes the fate and function of the phages themselves. Such insights add another layer of complexity to phage therapy but also reveal new therapeutic opportunities. By understanding and manipulating this dynamic, phages might be used not only to eliminate harmful bacteria but also to recalibrate immune responses in a way that benefits long-term host health.

The potential of phages to dynamically interact with the host immune system, reshaping both local and systemic immune responses, positions them as valuable tools in precision medicine. Future studies are needed to further elucidate the mechanisms under-lying phage-cytokine interactions, optimize phage engineering techniques, and ensure safety and efficacy in clinical settings [22].

## 7. Challenges and Future Directions

Despite the promising evidence supporting the therapeutic potential of phage-based cytokine modulation, significant challenges must be addressed before these strategies can be reliably translated into clinical practice. One of the primary hurdles is the need for further research to fully elucidate the complex molecular interactions between bacteriophages and cytokine signaling pathways. While some studies have begun to reveal the broad effects that phages can have on immune responses, much remains unknown about the specific mechanisms at play. A deeper understanding of these interactions at the molecular level is critical for designing phage-based therapies that can precisely modulate immune responses in a predictable and controlled manner. This knowledge will also allow scientists to better tailor phage therapies for specific conditions, enhancing their therapeutic efficacy and safety [126].

In addition to these scientific challenges, there are significant regulatory and safety concerns that must be carefully addressed before phage-based therapies can be introduced into clinical settings. The use of engineered phages in human patients requires stringent regulatory oversight to ensure that the therapies meet safety and efficacy standards. These standards must be established with an eye toward both the short- and long-term impacts of phage therapy, including potential side effects or unintended interactions with the host’s immune system. The regulatory process for phage therapy is particularly challenging due to the complexity of these agents, which are biologically distinct from traditional pharmaceuticals. Therefore, it is crucial to develop clear guidelines for the approval of phage-based treatments, considering both their unique properties and their potential to serve as living organisms that can evolve over time. Furthermore, establishing rigorous quality control measures for phage preparations is vital to ensure consistency between different therapeutic batches. This includes confirming the purity, potency, and stability of phage preparations, which is crucial for the reproducibility and safety of treatments [127].

Another considerable challenge lies in the inherent variability of the human microbiome and immune responses. The microbiome, which plays a central role in regulating immune function, differs widely between individuals and can significantly influence the outcomes of phage therapy. This variability poses a challenge for designing universal phage-based therapies that work across all patient populations. For example, individual differences in microbiome composition could lead to divergent responses to phage therapy, either enhancing or hindering the desired immune modulation. Similarly, patients with varying immune system responses might experience different therapeutic outcomes. As a result, phage therapies may need to be personalized to account for these individual differences. To address this challenge, large-scale, longitudinal studies are needed to better understand how phages interact with the microbiome and how these interactions might influence immune responses in different individuals. Such studies could help identify biomarkers that predict therapeutic success and allow for more precise targeting of phage therapies to those most likely to benefit [123].

Furthermore, the potential for off-target effects and unintended consequences of phage-based therapies warrants careful consideration. While phages are highly specific to their bacterial hosts, the possibility of cross-reactivity or unforeseen immune responses cannot be ruled out. Research into phage–host interactions must prioritize the identification of potential risks, such as the development of resistance, immune evasion strategies, or negative impacts on the host microbiome. Addressing these concerns will require a multidisciplinary approach, integrating expertise from microbiology, immunology, and pharmacology, to ensure that phage-based therapies are both safe and effective.

## 8. Conclusions

The interactions between bacteriophages and cytokines represent an exciting and rapidly evolving area of research, with the potential to revolutionize our approach to immunotherapy and infectious disease treatment. While the field is still in its infancy, emerging evidence suggests that phages can influence not only bacterial populations but also immune responses in profound ways. The ability of phages to modulate cytokine networks within the immune system could provide novel therapeutic strategies for treating infections, autoimmune diseases, and even cancer. This expanding understanding of phage biology marks a paradigmatic shift in our thinking, revealing phages as more than just bacterial predators. Rather, they are poised to be harnessed as powerful, dynamic agents capable of fine-tuning immune function and restoring balance to dysregulated systems.

As this research progresses, future studies must prioritize a deeper exploration of specific phage–host interactions. Identifying the precise mechanisms through which phages influence cytokine production and immune cell behavior will be essential for translating these findings into clinical applications. For example, understanding how phages interact with immune cells such as dendritic cells, macrophages, and T cells could reveal new insights into how to enhance the immune response against infections or malignancies. Similarly, characterizing the cytokine pathways that are modulated by phages will allow researchers to pinpoint key regulatory networks that could be targeted for therapeutic purposes.

Beyond basic research, there is also a need for the development of targeted phage-based immunotherapies. These therapies should be designed with the goal of selectively modulating specific immune responses in a controlled and predictable manner, minimizing the risk of unintended consequences. This will likely require the engineering of phages with tailored properties, such as the ability to deliver cytokines directly to immune cells or to activate specific immune pathways while avoiding off-target effects. Additionally, integrating phage therapy with other forms of immunotherapy, such as checkpoint inhibitors or monoclonal antibodies, could provide synergistic benefits, offering a more comprehensive approach to treating complex diseases like cancer and chronic infections.

In conclusion, bacteriophages represent an auspicious frontier in the field of immunotherapy. By exploiting their ability to modulate immune responses, we could open new avenues for treating a wide range of diseases. As the body of research in this area continues to grow, the potential for phage-based therapies to alter our approach to infection control, immune regulation, and even cancer treatment becomes increasingly concrete. However, significant work remains to address the scientific, regulatory, and clinical challenges that stand in the way of their widespread use. With continued innovation and interdisciplinary collaboration, phages may one day become a cornerstone of modern immunotherapy, offering new hope for patients around the world.

## Figures and Tables

**Figure 1 biomedicines-13-01202-f001:**
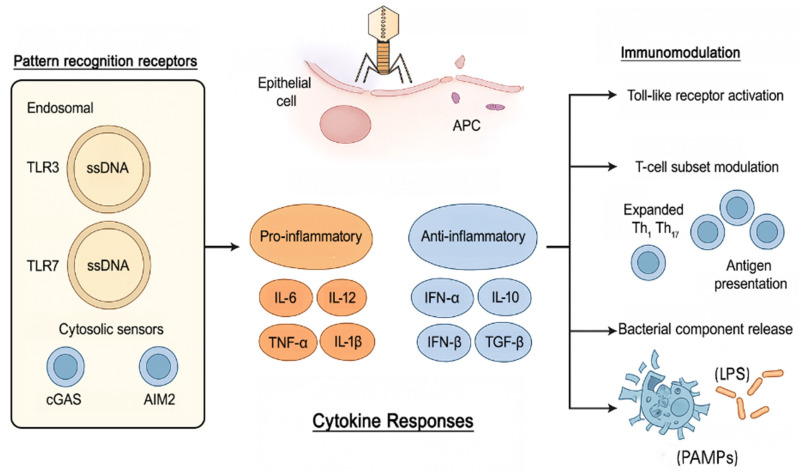
Mechanisms of immune modulation by bacteriophages. Phages interact with immune receptors leading to cytokine responses: pro-inflammatory (IL-6, IL-12, IL-1β) and anti-inflammatory (IFN-α, IFN-β, IL-10, TGF-β). Immunomodulatory mechanisms include receptor activation, T-cell subset modulation (Th1, Th17), and bacterial component release (LPS, peptidoglycan) affecting immune homeostasis.

**Figure 2 biomedicines-13-01202-f002:**
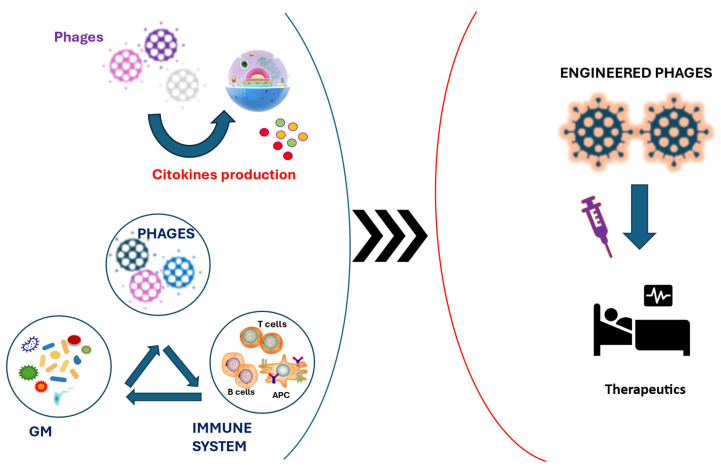
Potential therapeutic applications of phages, gut microbiota, and immune system interactions. Phages regulate immune responses through cytokine production and direct interactions with immune cells, while establishing dynamic crosstalk with gut microbiota. Promisingly, recent advances in phage engineering allow the creation of engineered phage therapy strategies, thus expanding tools available for precision medicine. Abbreviation: GM: gut microbiota.

## Data Availability

No new data were created or analyzed in this study.

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
