# Peer review of "Cytokines Meet Phages: A Revolutionary Pathway to Modulating Immunity and Microbial Balance"

_biomedicines, 2025, doi:10.3390/biomedicines13051202_

Round 1

Reviewer 1 Report

Comments and Suggestions for Authors

Topic: Cytokines Meet Phages: A Revolutionary Pathway to Modulating Immunity and Microbial Balance

The manuscript is well-written and easy to follow. The topic is highly relevant and intriguing, especially considering the growing interest in the intersection between bacteriophage therapy and immune modulation. The authors acknowledge several current challenges in the field, including the limited understanding of specific mechanisms of action, as well as unresolved issues related to regulation and safety of phage-based therapies. Nonetheless, this review represents a timely and valuable first step in consolidating recent findings on the interactions between phages and cytokines.

Comments:

  1. The review is largely narrative in nature and currently includes only a single figure. To enhance clarity and reader engagement, it would be beneficial to incorporate additional schematic illustrations—particularly those summarizing the mechanisms of immune modulation by phages.

  2. Given that this is a review article, each sub-topic could benefit from a more in-depth discussion. I recommend that the authors expand on each section by incorporating additional recent findings and mechanistic insights, which would provide a more comprehensive and informative resource for readers.

Author Response

Dear Editor of Biomedicines 

First, my coauthors and I would like to thank you sincerely for this opportunity to cooperate. We profoundly thank the reviewers for the comments and useful suggestions to improve the paper. We thank You for your constructive critique and hope the review process has improved the manuscript. If additional changes are warranted, we will make them.  

We hope that this revised version of our manuscript may now be found suitable for publication.  

This is a point-by-point list of changes made in the paper:

Reviewer 1

The manuscript is well-written and easy to follow. The topic is highly relevant and intriguing, especially considering the growing interest in the intersection between bacteriophage therapy and immune modulation. The authors acknowledge several current challenges in the field, including the limited understanding of specific mechanisms of action, as well as unresolved issues related to regulation and safety of phage-based therapies. Nonetheless, this review represents a timely and valuable first step in consolidating recent findings on the interactions between phages and cytokines.

Comments:

  1. The review is largely narrative in nature and currently includes only a single figure. To enhance clarity and reader engagement, it would be beneficial to incorporate additional schematic illustrations—particularly those summarizing the mechanisms of immune modulation by phages.

We thank you for your valuable comment. We added the figure 1 that summarizes the mechanisms of immune modulation by phages.

  1. Given that this is a review article, each sub-topic could benefit from a more in-depth discussion. I recommend that the authors expand on each section by incorporating additional recent findings and mechanistic insights, which would provide a more comprehensive and informative resource for readers.
    We thank you for your valuable comment. We received the suggestion that each section of the manuscript be expanded upon. In this revised version, we have expanded the various subsections to include recent studies and further insights into the mechanisms involved.

We thank You for your constructive critique and we hope the review process has led to an improved manuscript. 

If additional changes are warranted, we will make them. 

We hope that this revised version of our manuscript may now be found suitable for publication. 

Sincerely,

Rossella Cianci

Reviewer 2 Report

Comments and Suggestions for Authors

A good and interesting review on the potential of phages as therapeutic tools in immunity and infection. Although the subject is still not developed the authors examin the possibilities of using phages in medicine. The review is a little too long and I suggest to shorten/delete the following parts:

Introduction: not needed all the historical part

line 276: please explain how phages can recognize cancer cells

lines 391-398 is a repetition of what said before

lines 424-438 can be eleminated because repeat text in the previous pages

Author Response

Dear Editor of Biomedicines 

First, my coauthors and I would like to thank you sincerely for this opportunity to cooperate. We profoundly thank the reviewers for the comments and useful suggestions to improve the paper. We thank You for your constructive critique and hope the review process has improved the manuscript. If additional changes are warranted, we will make them.  

We hope that this revised version of our manuscript may now be found suitable for publication.  

This is a point-by-point list of changes made in the paper:

Reviewer 2

A good and interesting review on the potential of phages as therapeutic tools in immunity and infection. Although the subject is still not developed the authors examin the possibilities of using phages in medicine. The review is a little too long and I suggest to shorten/delete the following parts:

  • Introduction: not needed all the historical part
  • line 276: please explain how phages can recognize cancer cells
  • lines 391-398 is a repetition of what said before
  • lines 424-438 can be eleminated because repeat text in the previous pages

We have made the requested changes

We thank You for your constructive critique and we hope the review process has led to an improved manuscript. 

If additional changes are warranted, we will make them. 

We hope that this revised version of our manuscript may now be found suitable for publication. 

Sincerely,

Rossella Cianci